

# Mechanical nociceptive assessment of the equine hoof following distal interphalangeal joint intra-articular anesthesia

Bruno D. Malacarne[1], Leticia O. Cota[1], Antônio C.P. Neto[1], Cahuê F.R. Paz[2], Lucas A. Dias[1], Mayara G. Corrêa[1], Armando M. Carvalho[1], Rafael R. Faleiros[1,3] and Andressa B.S. Xavier[1]

[1] Equinova Research Group –Department of Veterinary Clinics and Surgery, School of Veterinary, Universidade Federal de Minas Gerais, Belo Horizonte, Minas Gerais, Brazil
[2] Centro Universitário INTA-Uninta, Sobral, Ceará, Brazil
[3] Conselho Nacional de Desenvolvimento Científico e Tecnològico –CNPq, Brasília, Brazil

## ABSTRACT

**Background**. With the hypothesis that equine dorsal lamellar tissue can be desensitized by anesthesia injection into distal interphalangeal joint (DIPJ), the objective was to assess the mechanical nociceptive threshold of hoof dorsal lamellae following intra-articular (IA) administration of lidocaine into this joint.

**Methods**. The DIPJ of the forelimbs of six adult healthy horses were injected with either 5 mL of lidocaine, or 5 mL of lactated Ringer's solution. Treatments were randomly distributed, with each forelimb undergoing a single treatment. The hooves were evaluated pre- and post-injection at pre-selected times over 4 h, using a pressure algometry model. Mechanical nociceptive thresholds (MNTs) were recorded for the sole (dorsal, palmarolateral, and palmaromedial regions), coronary band (medial, lateral, and dorsal regions), heel bulbs (medial and lateral), and dorsal lamellar region (2 cm and 4 cm distal to the coronary band). The MNT means were compared over time using the Friedman test and between treatments using the Wilcoxon signed-rank test, with values of $P < 0.05$ considered statistically significant.

**Results**. There were no differences between treatments for any region of the hoof during the evaluation period. However, MNT values indicating analgesia were recorded in the dorsal lamellar region in 50% of hooves following adminstration of lidocaine into the DIPJ.

**Conclusion**. The administration of 5 mL of lidocaine into the DIPJ does not significantly increase the mechanical nociceptive threshold of the equine hoof.

Corresponding author
Armando M. Carvalho,
armandodvm@gmail.com

## INTRODUCTION

To date, diagnostic anesthesia remains the best method for localization of lameness pain in horses, although misinterpretation due unspecific analgesia may occur (*Schumacher & Schramme, 2018*). The lack of specificity of the distal interphalangeal joint (DIPJ)

anesthetic block is still a focus of discussion, due to evidences of local drug diffusion into periarticular tissues such the navicular bursa (*Gough, Mayhew & Munroe, 2002*) and the navicular bone (*Keegan et al., 1996*). Anesthetics injected into the DIPJ are thought to diffuse through tissues in defined sequences; therefore, frequent evaluations post-injection are considered important for determining if the correct anatomical location has been affected (*Bowker, 2007*). Evaluation and interpretation of the analgesic effect on the DIPJ or navicular bursa should occur approximately 5–10 min post-injection (*Schumacher et al., 2013*). Reportedly, intra-articular (IA) administration of mepivacaine into the DIPJ desensitizes the DIPJ (*Easter et al., 2000*), navicular bursa (*Pleasant et al., 1997*; *Gough, Mayhew & Munroe, 2002*), navicular bone (*Dyson & Kidd, 1993*), collateral ligaments of the navicular bone (*Bowker et al., 1997*), dorsal and palmar sole region, depending on the volume of mepivacaine administered (*Schumacher et al., 2000*; *Schumacher et al., 2001*), and distal portion of the deep digital flexor tendon (DDFT) (*Dyson et al., 2003*). With the hypothesis that the dorsal lamellar tissue of the equine hoof can be desensitized by local anesthesia of the DIPJ, the objective of this study was to use pressure algometry to evaluate the mechanical nociceptive threshold (MNT) in pre-selected regions of the equine hoof following IA administration of 5 mL of 2% lidocaine into the DIPJ.

## MATERIALS & METHODS

### Animals and experimental design
The experiment was approved by the Ethics Committee on Animal Use of Universidade Federal de Minas Gerais (UFMG) (Approval No.360, February 27, 2018). Six mixed-breed horses of varying sex and neuter status (4 mares, 1 gelding, and 1 stallion) were used in randomized blocks and repeated measurements over time design. The horses were 7–13 years of age (median, 9 years), 142–164 cm tall (median, 146 cm), and weighed 350–460 kg (median, 355 kg). All horses were considered clinically healthy, determined via physical and orthopedic examination. A complete lameness examination was also performed on all horses to assure that no lameness was present prior to beginning the study (*Kaneps, 2014*).

### Hoof preparation
Prior to the start of the study, horses were restrained in stocks, and their hooves were cleaned and trimmed for geometric balance. Hoof preparation for pressure algometry testing was performed as previously described by *Paz et al. (2016)*. Horses were sedated with detomidine (20 $\mu$g kg$^{-1}$ intravenous (IV); Detomidin, Sintec) followed by intramuscular (IM) administration of tetanus antitoxin (5000 IU; Lema Injex) and aseptic preparation of the hoof capsule. Two hoof wall defects were made in the center of the dorsal hoof wall of each fore foot, using a 10 mm drill bit mounted onto a drill (Dewalt 12v max). The first defect was located 2 cm distal to the coronary band and the second 4 cm distal to the coronary band. Hoof wall defects penetrated only the insensitive horn, stopping immediately before reaching the sensitive lamellae. Using a hoof knife, the sole of each hoof was prepared by removing the keratinized layer until reaching the sensitive tissue near

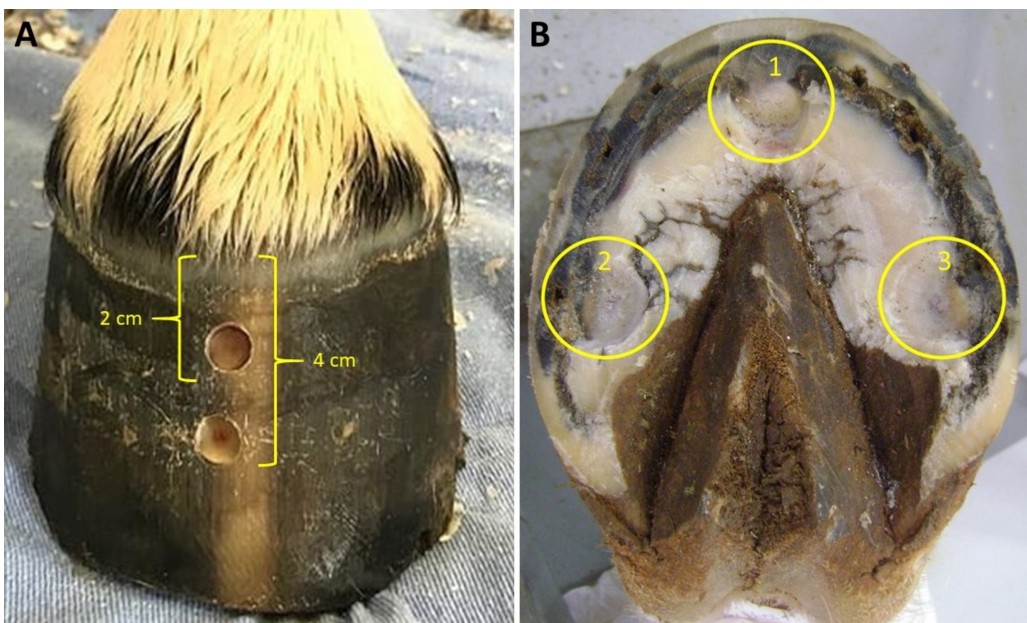

**Figure 1** **Equine hoof capsule images showing the points for mechanical nociceptive threshold assessment in the dorsal hoof wall and sole.** (A) Defects created in the dorsal hoof wall, 2 and 4 centimeters away from the coronary band. (B) Defects created at dorsal (1), palmaromedial (2) and palmarolateral (3) borders of sole. Note that horn was removed in order to reach the flexible portion of the cornified tissue, but without causing exposure or injury of the underneath corium.

the solar corium at three different points close to the palmarolateral, palmaromedial, and dorsal borders of the sole (Fig. 1).

## Injection of the DIPJ

Injections into the DIPJs were performed 24 h following preparation of the hooves and in complete absence of any sedative effects. Treatments were randomized in each horse, with one DIPJ receiving 5 mL of 2% lidocaine (Xylestesin 2%, Laboratory Cristália Pharmaceuticals Ltd) and the other DIPJ receiving 5 mL of lactated Ringer's solution (Samtec Biotechnology). The horses were restrained in stocks, and a nose twitch was applied. Complete aseptic preparation of the the injection site was performed prior to administering the treatment into each forelimb. A dorsolateral DIPJ injection technique was used, as previously described (*Moyer, Schumacher & Schumacher, 2011*). The presence of synovial fluid in the hub of the needle and the absence of resistance during injection confirmed access into the joint. Either lactated Ringer's solution or 2% lidocaine was administered through a 30 mm × 0.8 mm needle and 5 mL syringe.

## Algometry

Pressure algometry was utilized to evaluate the MNT for each hoof. A portable dynamometer (Instrutemp 20kgf ITFG-5020), calibrated by the manufacturer for compression, was used to determined the force (kg) necessary to incite a response. Using a previously described technique (*Haussler, Behre & Hill, 2008*), the dynamometer was

applied with a constant increase in pressure at a 90° angle in relation to the contact surface. Pressure algometry was performed on each hoof at 10 different sites at the following locations: coronary band (medial, lateral, dorsal regions), heel bulbs (medial and lateral), sole (dorsal, palmaromedial, and palmarolateral regions), and dorsal lamellae (2 cm and 4 cm distal to the coronary band). All limbs are loaded during assessements, with exception of the sole sites. A 7 mm diameter flat tip was used for testing of the coronary band and heel bulbs, whereas a cone tip was used to evaluate the dorsal lamellae and sole regions (*Paz et al., 2016*).

To avoid environmental distractions and visual perception of the algometer being applied, a blindfold was applied over the eyes of each horse during each evaluation. The basal MNT for each foot was determined prior to IA administration of each treatment. The minimum force required to stimulate withdrawal of the limb was recorded (*Zarucco et al., 2010*). Sites were assessed at 13 pre-selected times, beginning 10 min pre-injection (baseline), followed by 5, 10, 15, 20, 30, 60, 90, 120, 150, 180, 210, and 240 min post-injection. To avoid trauma to the horse's foot, a maximum force of 6 kg was set to determine responsiveness based on previously studies, which showed that horses did not react when forces over this value were applied (*Zarucco et al., 2010*; *Paz et al., 2016*).

Preparation of the injectate was performed by an individual not blinded to the treatments. Joint injections and dynamometer measurements were performed by a single investigator blinded to the randomized treatments. Two additional evaluators, also blinded to the treatments administered, evaluated the horses response to the applied instrument pressure. The response was determined twice at each time in intervals of 2–4 s, and the average of the two readings was obtained. When divergence in a reading occurred between the evaluators, the value was disregarded and the procedure repeated. After the experimental period, all horses received phenylbutazone (4.4 mg kg$^{-1}$; Ourofino) IV daily for 5 days.

## Statistical methods

The data were analized by GraphPad Prism 7, and tested for normality using the Shapiro–Wilk and Kolmogorov–Smirnov tests and determined to be not normally distributed. Therefore, the data were tested non-parametrically using the Friedman test. Post hoc comparisons were made using Dunn's test for comparison between times within each group and the Wilcoxon signed-rank test for comparison between groups within each time. For all analyses, $P < 0.05$ was considered significant.

## RESULTS

The MNT for each location tested are represented in Figs. 2–5. The results for the 150, 180, 210, and 240 min post-injection readings were nearly identical to the 120 min post-injection reading; therefore, only the 120 min value is represented graphically. There were no statistical differences ($P < 0.05$) over time or between treatment groups (2% lidocaine and lactated ringer's solution) for any of the variables analyzed (Table 1). However, signs of analgesia (MNT $\geq$ 3 kg) were observed at various times in the dorsal lamellar region in 3/6 (50%) of the horses studied (Fig. 2). In horse 1, the maximum force (6 kg) was applied 60 min post-injection in the lamellar region 4 cm distal the coronary band (Fig. 2). Horse

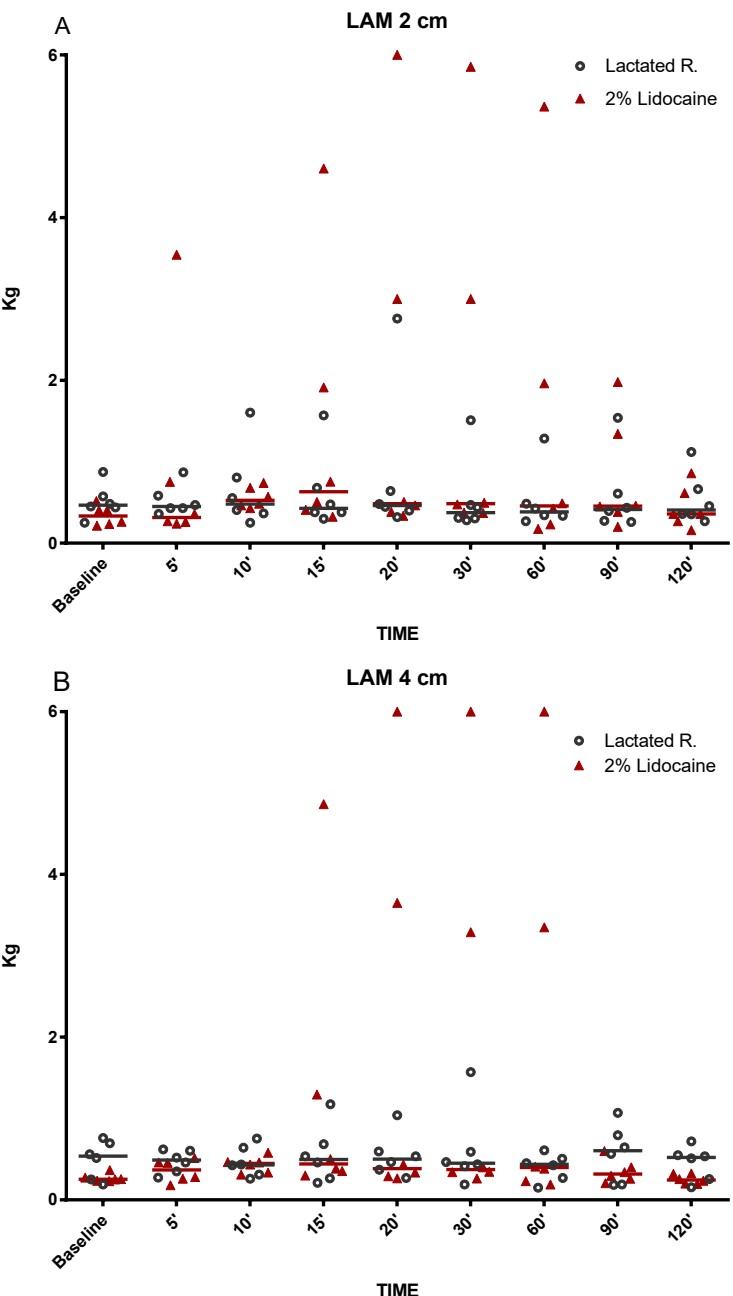

**Figure 2** **Graphical representation of mechanical nociceptive threshold values.** Evaluation of dorsal lamella regions (LAM). (A) LAM 2 cm and (B) LAM 4 cm distal to coronary band of hooves in healthy equine limbs, injected into distal interphalangeal joint with lactated Ringer's solution or 2% lidocaine. Each marker represent a single horse, median expressed in lines.

5 had MNT values $\geq$ 3 kg in the lamellar region 2 cm distal the coronary band at 15 min (4.60 kg), 20 min (3 kg), and 30 min (3 kg) and 4 cm distal to the coronary band at 15 min (4.80 kg), 20 min (6 kg), and 30 min (6 kg). Horse 6 had an MNT $\geq$ 3 kg in the lamellar region 2 cm distal the coronary band at 20 min (6 kg), 30 min (5.85 kg), and 60 min (5.36

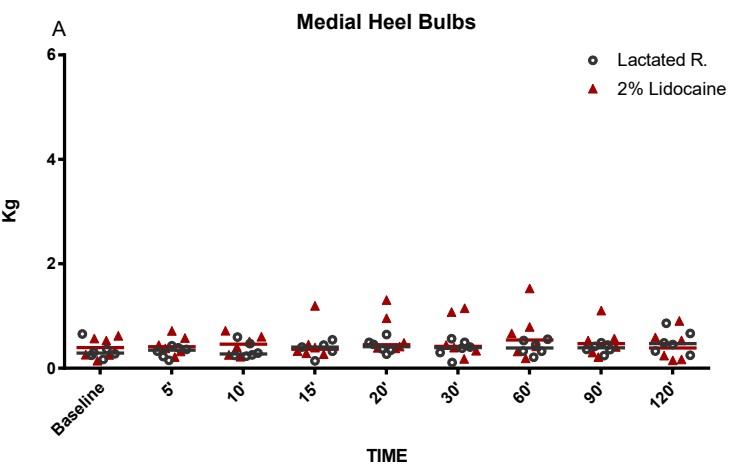

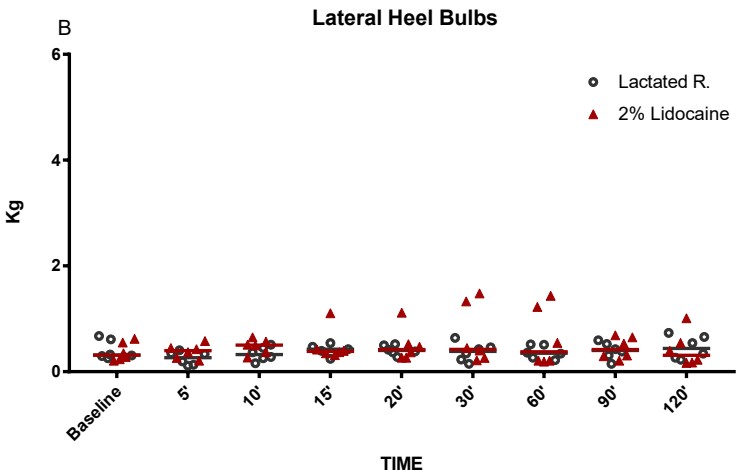

**Figure 3  Graphical representation of mechanical nociceptive threshold values.** Evaluation of heel bulbs (A) medial and (B) lateral in healthy equine limbs, injected into distal interphalangeal joint with lactated Ringer's solution or 2% lidocaine. Each marker represent a single horse, median expressed in lines.

kg) and 4 cm distal to the coronary band at 20 min (3.65 kg), 30 min (3.29 kg), and 60 min (3.35 kg) (Fig. 2). Only one horse had an MNT > 3 kg in the palmaromedial region of the sole, tolerating 6 kg at 60 min post-injection.

## DISCUSSION

This study demonstrates that IA injection of the DIPJ with 5 mL of lidocaine does not result in statistical increases ($P < 0.05$) in MNT within the dorsal lamellae, sole, coronary band, or heel bulbs. However, 3/6 horses in this study demonstrated partial desensitization in the dorsal lamellar region between 15 and 60 min post-injection. These results suggest that diffusion of lidocaine from the DIPJ to the innervation of the dorsal lamellae should

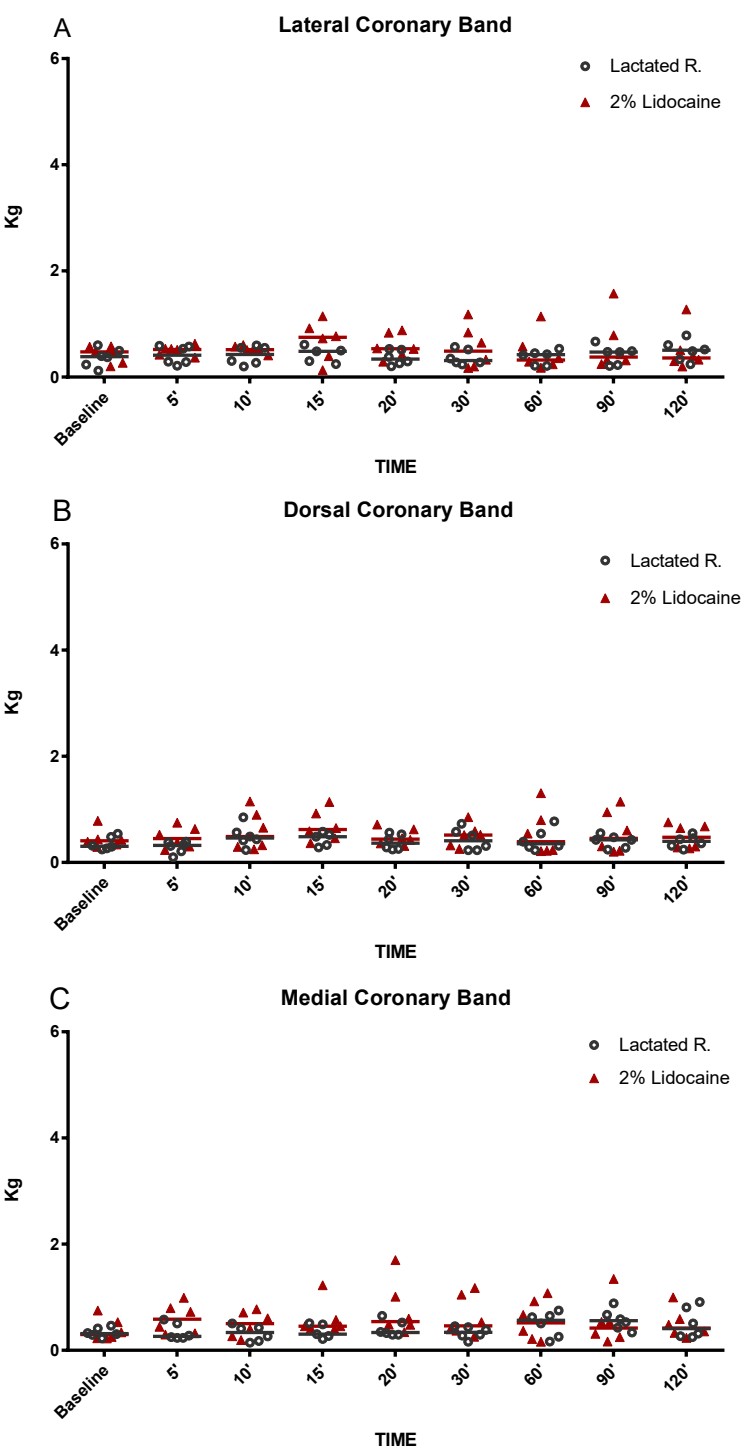

**Figure 4** **Graphical representation of mechanical nociceptive threshold values.** Evaluation of coronary band (A) lateral, (B) dorsal and (C) medial in healthy equine limbs, injected into distal interphalangeal joint with lactated Ringer's solution or 2% lidocaine. Each marker represent a single horse, median expressed in lines.

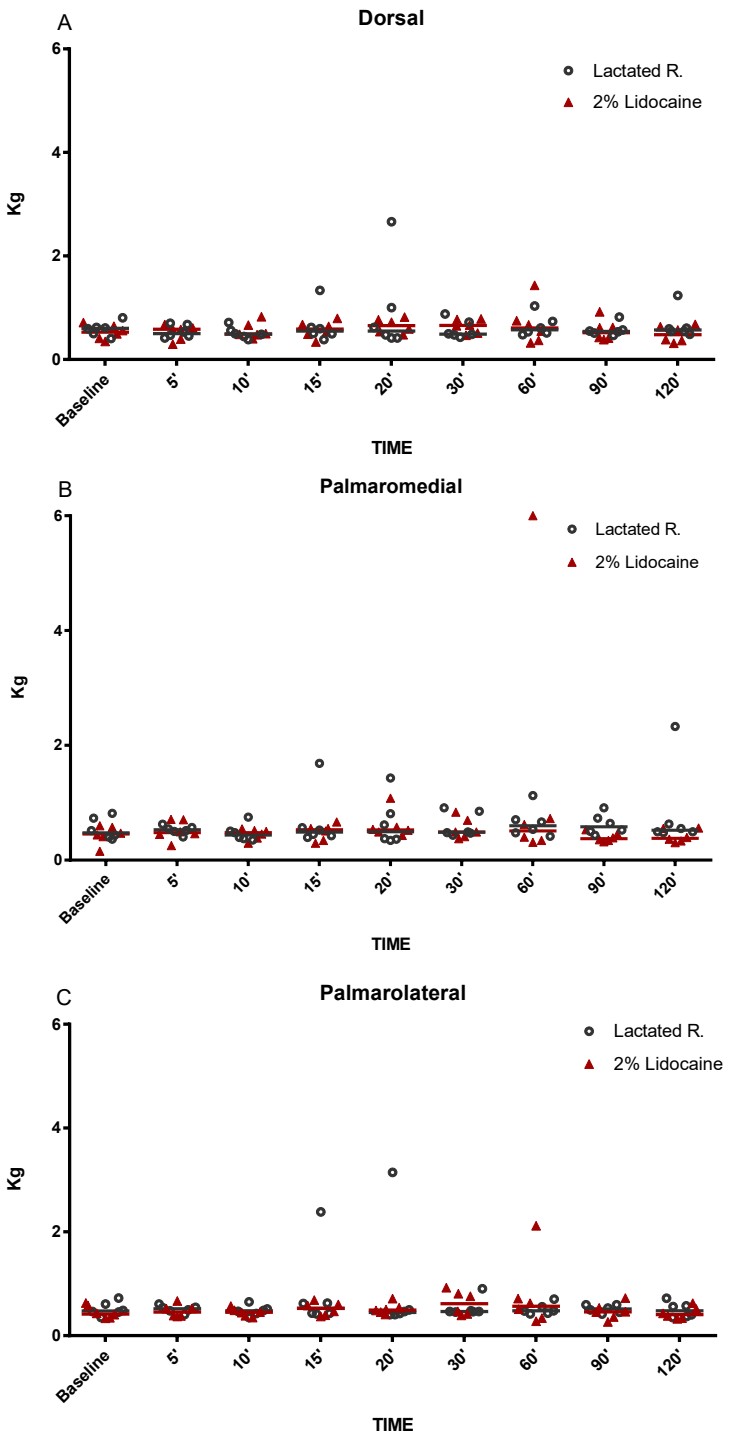

**Figure 5  Graphical representation of mechanical nociceptive threshold values.** Evaluation of sole borders (A) dorsal, (B) palmaromedial and (C) palmarolateral in healthy equine limbs, injected into distal interphalangeal joint with lactated Ringer's solution or 2% lidocaine. Each marker represent a single horse, median expressed in lines.

be considered when interpreting the effects of a DIPJ block. For diagnostic purposes, it is recommended to use the minimum anesthetic volume necessary for effective desensitization (*Schumacher & Schramme, 2018*). *Moyer, Schumacher & Schumacher (2011)* advocates using a volume of 4–6 mL for anesthesia of the DIPJ. *Dyson (1998)* suggests that the volume administered should not exceed 6 mL, to avoid distention of the joint and leaking of the local anesthetic. Excessive distention of the joint with an anesthetic agent may promote subcutaneous extravasation at the injection site, increased diffusion, and desensitization of unintended structures (*Schumacher & Schramme, 2018*). Consequently, low volumes of anesthetics may improve the specificity of DIPJ blocks; therefore, low volumes were used in this study. Despite using a low volume (5 mL) of lidocaine, 50% of horses in this study demonstrated desensitization of the dorsal lamellae. Desensitization of the dorsal lamellae via perineural block of the palmar digital nerve (PDN) has been reported by *Paz et al. (2016)*, determining that this region is innervated mainly by the medial and lateral PDNs. However, the possibility of the dorsal nerve branches contributing to dorsal lamellar tissue innervation in some horses was suggested in that study. Findings from our study suggest that, even with low volumes of lidocaine, injection into the DIPJ can result in anesthesia of the PDNs and/or their dorsal branches in some horses.

While some horses demonstrated anesthesia of the dorsal lamellae after administration of lidocaine into the DIPJ, there was no statistical difference between treatment groups. This could be explained by the variability in MNT responses in addition to a small sample size. Variability in MNT responses between individuals has been observed when evaluating back pain in horses (*Haussler & Erb, 2006*). However, pressure algometry has been used reliably to identify and quantify MNT within the equine digit, particularly for evaluation of coronary band (*Zarucco et al., 2010*), heel bulbs (*Jordana et al., 2014*), dorsal lamellar region, and sole (*Paz et al., 2016*) desensitization after local anesthestic administration. Furthermore, each horse was evaluated prior to enrollment in the study to minimize variability. Horses were not enrolled into the study if scarring of the distal limbs was noted or if they exhibited MNT responses >1 kg of force. Different from that, previous studies have reported MNT over 1 kg of force in clinically normal horses (*Haussler & Erb, 2006*). Many factors may have contributed to such divergences like breed, level of horsemanship, part of the body, hair coat, and rate of pressure application of the probe, but the major cause seems to be the probe configuration (*Taylor et al., 2016*). *Haussler & Erb (2006)* used a probe cover with rubber. Metallic probes were preferred for the current hoof model based on conclusions from *Taylor et al. (2016)* that smaller probe tips may be preferable as MNT data are less variable. In fact, baseline values registered in both groups were similar from those previously reported using metallic probes with flat and conical tips (*Taylor et al., 2016*). Three of the six horses had MNTs of 3 kg and 6 kg, occurring between 15 and 60 min post-injection of lidocaine. Anesthesia of the dorsal lamellae could be explained by either inadvertent infiltration of the subcutaneous tissues with lidocaine during IA injection or by diffusion of lidocaine from the DIPJ to other tissues. However, the explanation that diffusion occurred seems more likely, as MNT values ≥3 kg occurred after 15 min. Previous studies have reported that anesthetic diffusion to the palmar structures of the foot occurred at a similar time post-injection, determined by improvement of lameness

**Table 1  Means (range) mechanical nociceptive thresholds (in kg) obtained from the dorsal lamella (LAM, 2 and 4 cm distal to the coronary band), sole, bulbs of the heel and coronary band of the forelimb hooves in horses subjected to distal interphalangeal joint infiltration with 5 mL of lidocaine 2% (L) or Ringer's lactated solution (RS).**

| Treatment-Region | Baseline | Minutes | | | | | | | | |
|---|---|---|---|---|---|---|---|---|---|---|
| | | 5′ | 10′ | 15′ | 20′ | 30′ | 60′ | 90′ | 120′ | |
| (RS)-LAM 2cm | 0.51 (0.25–0.87) | 0.52 (0.36–0.87) | 0.66 (0.25–1.60) | 0.63 (0.30–1.57) | 0.84 (0.32–2.76) | 0.55 (0.28–1.51) | 0.52 (0.27–1.28) | 0.58 (0.26–1.54) | 0.53 (0.27–1.12) | |
| (L)-LAM 2cm | 0.34 (0.21–0.51) | 0.90 (0.24–3.54) | 0.56 (0.43–0.74) | 1.41 (0.32–4.60) | 1.78 (0.33–3.75) | 1.76 (0.37–5.85) | 1.44 (0.17–5.36) | 0.80 (0.2–1.98) | 0.49 (0.14–1.51) | |
| (RS)-LAM 4cm | 0.49 (0.19–0.76) | 0.47 (0.27–0.62) | 0.47 (0.26–0.75) | 0.55 (0.21–1.17) | 0.54 (0.27–1.04) | 0.61 (0.19–1.57) | 0.40 (0.15–0.61) | 0.57 (0.18–1.07) | 0.45 (0.15–0.72) | |
| (L)-LAM 4cm | 0.26 (0.23–0.36) | 0.35 (0.18–0.51) | 0.43 (0.31–0.58) | 1.28 (0.30–4.86) | 1.82 (0.26–6.00) | 1.77 (0.26–6.00) | 1.76 (0.19–6.00) | 0.35 (0.20–0.60) | 0.25 (0.19–0.32) | |
| (RS)-Sole (Dorsal) | 0.58 (0.40–0.80) | 0.54 (0.41–0.70) | 0.51 (0.38–0.71) | 0.65 (0.38–1.33) | 0.93 (0.41–2.66) | 0.58 (0.43–0.88) | 0.65 (0.48–1.03) | 0.57 (0.46–0.82) | 0.66 (0.48–1.23) | |
| (L)-Sole (Dorsal) | 0.52 (0.35–0.71) | 0.52 (0.29–0.67) | 0.55 (0.40–0.82) | 0.57 (0.34–0.79) | 0.65 (0.48–0.82) | 0.64 (0.47–0.78) | 0.68 (0.32–1.43) | 0.56 (0.38–0.92) | 0.49 (0.31–0.68) | |
| (RS)-Sole (Palmaromedial) | 0.54 (0.36–0.81) | 0.52 (0.40–0.62) | 0.47 (0.35–0.75) | 0.67 (0.39–1.68) | 0.65 (0.34–1.43) | 0.60 (0.43–0.91) | 0.65 (0.41–1.25) | 0.62 (0.43–0.91) | 0.83 (0.48–2.33) | |
| (L)-Sole (Palmaromedial) | 0.44 (0.15–0.60) | 0.51 (0.25–0.71) | 0.44 (0.29–0.54) | 0.48 (0.29–0.66) | 0.60 (0.43–1.08) | 0.54 (0.37–0.83) | 1.40 (0.31–6.00) | 0.39 (0.32–0.53) | 0.41 (0.30–0.56) | |
| (RS)-Sole (Palmarolateral) | 0.51 (0.35–0.72) | 0.51 (0.40–0.61) | 0.48 (0.35–0.65) | 0.81 (0.42–2.38) | 0.89 (0.40–3.14) | 0.53 (0.45–0.90) | 0.51 (0.42–0.70) | 0.51 (0.41–0.59) | 0.49 (0.35–0.32) | |
| (L)-Sole (Palmarolateral) | 0.45 (0.33–0.63) | 0.47 (0.37–0.67) | 0.44 (0.35–0.57) | 0.51 (0.37–0.68) | 0.51 (0.41–0.71) | 0.62 (0.39–0.92) | 0.76 (0.28–2.11) | 0.46 (0.26–0.72) | 0.43 (0.32–0.62) | |
| (RS)-Bulbs (Lateral) | 0.41 (0.25–0.67) | 0.25 (0.11–0.25) | 0.33 (0.16–0.51) | 0.41 (0.24–0.54) | 0.41 (0.28–0.52) | 0.37 (0.15–0.64) | 0.36 (0.22–0.51) | 0.39 (0.15–0.59) | 0.46 (0.22–0.73) | |
| (L)-Bulbs (Lateral) | 0.37 (0.21–0.62) | 0.38 (0.21–0.58) | 0.47 (0.26–0.65) | 0.49 (0.31–1.10) | 0.50 (0.26–1.11) | 0.68 (0.21–1.48) | 0.63 (0.19–1.43) | 0.44 (0.21–0.69) | 0.41 (0.16–1.01) | |
| (RS)-Bulbs (Medial) | 0.33 (0.17–0.65) | 0.31 (0.15–0.43) | 0.34 (0.23–0.59) | 0.37 (0.14–0.54) | 0.43 (0.27–0.64) | 0.37 (0.11–0.56) | 0.39 (0.21–0.55) | 0.38 (0.24–0.49) | 0.50 (0.25–0.71) | |
| (L)-Bulbs (Medial) | 0.39 (0.14–0.62) | 0.44 (0.21–0.71) | 0.45 (0.22–0.71) | 0.48 (0.27–1.19) | 0.65 (0.38–1.30) | 0.59 (0.18–1.15) | 0.65 (0.19–1.53) | 0.52 (0.21–1.10) | 0.43 (0.15–0.90) | |
| (RS)-Coronary band (Lateral) | 0.36 (0.12–0.60) | 0.41 (0.21–0.59) | 0.41 (0.20–0.59) | 0.42 (0.24–0.61) | 0.36 (0.20–0.53) | 0.37 (0.23–0.57) | 0.37 (0.21–0.53) | 0.42 (0.21–0.67) | 0.49 (0.24–0.78) | |
| (L)-Coronary band (Lateral) | 0.42 (0.20–0.58) | 0.49 (0.36–0.62) | 0.52 (0.41–0.60) | 0.68 (0.13–1.14) | 0.58 (0.29–0.88) | 0.56 (0.17–1.18) | 0.46 (0.17–1.14) | 0.61 (0.24–1.57) | 0.50 (0.20–1.27) | |
| (RS)-Coronary band (Medial) | 0.33 (0.22–0.46) | 0.34 (0.23–0.58) | 0.32 (0.14–0.50) | 0.35 (0.21–0.51) | 0.40 (0.29–0.65) | 0.33 (0.16–0.45) | 0.49 (0.16–0.75) | 0.57 (0.33–0.88) | 0.50 (0.25–0.91) | |
| (L)-Coronary band (Medial) | 0.38 (0.22–0.75) | 0.59 (0.29–0.99) | 0.49 (0.19–0.77) | 0.58 (0.39–1.22) | 0.76 (0.34–1.70) | 0.62 (0.25–1.17) | 0.56 (0.16–1.07) | 0.52 (0.16–1.34) | 0.49 (0.23–0.99) | |
| (RS)-Coronary band (Dorsal) | 0.35 (0.24–0.54) | 0.28 (0.10–0.39) | 0.49 (0.23–0.85) | 0.44 (0.28–0.58) | 0.38 (0.24–0.56) | 0.43 (0.23–0.73) | 0.42 (0.23–0.77) | 0.39 (0.24–0.55) | 0.39 (0.24–0.55) | |
| (L)-Coronary band (Dorsal) | 0.44 (0.29–0.78) | 0.46 (0.23–0.75) | 0.59 (0.25–1.15) | 0.68 (0.36–1.14) | 0.48 (0.31–0.71) | 0.50 (0.25–0.85) | 0.55 (0.21–1.30) | 0.57 (0.20–1.14) | 0.48 (0.26–0.75) | |
associated with the podotrochlear apparatus after infusion of 5 ml of lidocaine into the DIPJ (*Dau et al., 2017*). Additionally, MNT values were <1.5 kg in the coronary band region at all evaluation times, suggesting that inadvertent infiltration of the subcutaneous tissues was not the source of the anesthetic effect at the dorsal lamellae. It is suspected that, despite the use of a low injection volume, diffusion of lidocaine from the joint occurred resulting in desensitization of the dorsal lamellar region via the dorsal branches of the PDNs. However, there is no evidence that the volume of anesthetic used in the present study was sufficient to diffuse into the palmar regions of the foot to the extent necessary for producing a complete perineural block of the PDNs. Previous studies have shown that perineural anesthesia of this nerve in the distal portion promotes consistent desensitization of the sole, heel bulbs, and dorsal lamellar tissue (*Paz et al., 2016*). Desensitization of the sole and heel bulbs did not occur in the present study. Therefore, if only the dorsal branches and lamellar terminations of the PDNs were desensitized by a low volume DIPJ block, an inconsistent MNT response at the dorsal lamellae may occur (*Paz et al., 2016*).

The solar region of the hoof can become desensitized following administration of an anesthetic agent into the DIPJ (*Schumacher et al., 2000*; *Schumacher et al., 2001*; *Sardari, Kazemi & Mohri, 2002*). Using an experimental model to induce lameness via sole pressure, horses were successfully desensitized in both the dorsal and palmar regions of the sole following injection of 10 mL of mepivacaine into the DIPJ (*Schumacher et al., 2000*). However, the use of a smaller injection volume (6 mL) promoted desensitization only in the dorsal region of the sole (*Schumacher et al., 2001*). In contrast, the present study using 5 mL of 2% lidocaine did not result in significant increases in MNT in any portion of the sole. The lack of significant desensitization of the solar region, demonstrated by response to pressure algometry, in these horses suggests that 5 mL of lidocaine is insufficient volume to cause diffusion to the branches of the PDNs that innervate the solar regions of the foot. Low injection volumes may not be large enough to result in overdistention of the palmar recess. For example, 3 mL of plastic polymer was not sufficient to completely fill the palmar recess of this joint (*Bowker et al., 1997*). Therefore, in the current study, it is unlikely that the DIPJ palmar pouch became overdistended by the administration of 5 mL of lidocaine, decreasing the possibility of anesthetic diffusion to the branches of the PDNs responsible for the innervation of the solar corium. In contrast to previous reports, horses in this experiment were confined to stocks during evaluations and not walked post-injection (*Schumacher et al., 2000*; *Schumacher et al., 2001*; *Sardari, Kazemi & Mohri, 2002*). *Gough, Mayhew & Munroe (2002)* suggests that the injection volume and resulting pressure within a synovial structure may affect the anesthetic diffusion rate into adjacent synovial structures. Therefore, the absence of joint movement may have resulted in lower IA pressure and, when combined with a low injection volume, made it difficult for lidocaine to diffuse into the surrounding tissues.

We also need to highlight that lidocaine was used in the present study and that most of the previous assays studying DIPJ anesthesia effects used mepivacaine, which could promote a different effect under the current set up/design. Another limitation of this study is the small number of horses and the lack of normal distribution of the data. Under such circuntances, data was analysed by nonparametric tests that has lower power than

do standard tests. By line of reason, a larger sample could indicate statistical difference between groups considering the MNT values of dorsal lamellae.

## CONCLUSIONS

Our findings suggest that the dorsal lamellae should be considered as a possible source of lameness pain following 2% lidocaine anesthetic block of the DIPJ in some horses. This information is potential relevant in face of increasing number of horses with subclinical signs of endocrinopathic laminitis reported nowadays (*Patterson-Kane, Karikoski & McGowan, 2018*). Further study and clinical observations are needed to confirm that local anesthetic may diffuse from the DIPJ affecting the invervation of the dorsal lamellar region.

## ACKNOWLEDGEMENTS

We would like to thank Dr. Britta Leise and Dr. Lee Ann Fugler for reviewing the manuscript.

### Funding

This work was supported by FAPEMIG, CNPq, CAPES, and the Office of the Dean for Research at Universidade Federal de Minas Gerais (PRPq-UFMG). The funders had no role in study design, data collection and analysis, decision to publish, or preparation of the manuscript.

### Grant Disclosures

The following grant information was disclosed by the authors:
FAPEMIG.
CNPq.
CAPES.
Office of the Dean for Research at Universidade Federal de Minas Gerais (PRPq-UFMG).

### Competing Interests

The authors declare there are no competing interests.

### Author Contributions

- Bruno D. Malacarne conceived and designed the experiments, performed the experiments, analyzed the data, prepared figures and/or tables, authored or reviewed drafts of the paper, and approved the final draft.
- Leticia O. Cota, Antônio C.P. Neto, Cahuê F.R. Paz, Lucas A. Dias and Mayara G. Corrêa performed the experiments, prepared figures and/or tables, and approved the final draft.
- Armando M. Carvalho, Rafael R. Faleiros and Andressa B.S. Xavier conceived and designed the experiments, analyzed the data, prepared figures and/or tables, authored or reviewed drafts of the paper, and approved the final draft.

## Animal Ethics

The following information was supplied relating to ethical approvals (i.e., approving body and any reference numbers):

Ethics Committee on Animal Use of the Universidade Federal de Minas Gerais approved this study (Approval No.360, February 27, 2018).

## Data Availability

The raw measurements are available as Supplemental File.

## Supplemental Information

Supplemental information for this article can be found online at http://dx.doi.org/10.7717/peerj.9469#supplemental-information.

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
