# Peer review of "Mechanical nociceptive assessment of the equine hoof following distal interphalangeal joint intra-articular anesthesia"

_PeerJ, doi:10.7717/peerj.9469_

## Round 0.1 · original submission · Major Revisions

Thank you very much for your submission to PeerJ. I am sorry that your article has been under review for a little while, but it has taken me longer than typical to find reviewers for your article. However, two experts in your field have now provided feedback on your article.

Both reviewers provided favorable reviews of your study and article. However, both requested a number of clarifications both with regards to your methods and results. They have both also asked for a more detailed and exhaustive discussion and I agree that that would be beneficial.

Please read each of the reviewers' comments carefully and provide point-by-point responses as you prepare your revised article for resubmission. I look forward to receiving your revised article.

·

Basic reporting

Overall, one of the best written papers that I have read in quite a while. Interesting application of pressure algometry to address an important clinical issue.

Experimental design

Line 45 – I am not sure that ‘dorsal sole’ is a correct anatomical description. To me, all aspects of the sole are on the palmar surface. Need to recheck to confirm appropriate terminology.

Line 75 – Need to clearly state if the limbs were loaded or unloaded for all measures.

Line 78 - How was 6 kg determined to be an appropriate level to prevent tissue injury?

Figure 1 – I am quite surprised that the majority of your MNT values are < 1 kg, hence do not provide any room for discrimination of values. It seems that the probe tip used was too severe in this model. If all values are < 1 kg then there is no real possibility of identifying any treatment effects (local anesthetic) given the intra-horse and inter-horse variability in MNT values.

Did you do pilot work to define the probe tip used in this model or was it borrowed from another research group? All of this needs to be added to the discussion as a possible factor for no significant differences.

I would also recommend doing power calculations to provide some guidance to future researchers using this model as to the sample size required to show significant results.

The legend also needs to let the reader know that each marker represents a single horse.

Figure 3 – Need to explain why the baseline coronary band MNT values are so low (<1 kg/cm2), compared to values reported in Haussler, Behre 2008 and others.

Need to add some discussion of the strengths and weaknesses of the algometry technique. It seems that you do not question the MNT values at all and attribute all changes to effects of the local anesthetic, which may or may not be true.

What is the reliable lowest range of values that the algometer used in this study can reliably detect? Is it sensitive enough to repeatably differentiate between 0.5 versus 0.7 kg/cm2, as an example?

Validity of the findings

No comment

Additional comments

No comment

Reviewer 2 ·

Basic reporting

No comment.

Experimental design

no comment.

Validity of the findings

no comment.

Additional comments

General comments to the authors:

This is an interesting manuscript which will merit publication. The information provided in this manuscript may be useful for veterinarians that deal with equine lameness and use lidocaine to perform these diagnostic blocks. Therefore, the authors must include in the discussion that the results of the study are valid only if LIDOCAINE is being used to block the distal interphalangeal joint. Most of the literature cited and used to contrast the results of the current study have used a long lasting local anesthetic solution (mepivacaine) which could have been a different effect under the described study set up/design. Please keep that in mind as you cite those papers.
Specific comments to the authors:

Abstract or summary: please remove “However, signs of analgesia in the dorsal lamellar tissue, beginning 15 minutes after administration, suggest that anesthetic diffusion may decrease the specificity of DIPJ anesthesia in some horses” as this was not demonstrated in the current study.

Line 3: Please clarify what you mean by “associated with errors”.

Line 4-8: Please reword to be more specific: there is no proven diffusion of local anesthetic solution from the distal interphalangeal joint into the periarticular tissues (if so please provide a reference). My understanding is that because the palmar [plantar]outpouching of the DIPJ is in close juxtaposition to the digital neurovascular bundles, this block can desensitize the digital neurovascular bundle and indirectly desensitize the extra-articular structures (i.e. the sole, navicular bone, etc).

Line 10: please define IA.

Line 17: please define MNT.

Line 32-43: please consider including images or illustrations of the hoof defects created for pressure algometry. These images will make readers understand your set up.

Lines 190-191: Your results do not support this statement so please reword. You stated in the first line of the discussion “This study demonstrates that IA injection of the DIPJ with 5 mL of lidocaine does not result in significant increases in MNTs within the dorsal lamellae, sole, coronary band, or heel bulb”.

Discussion: The authors must include in the discussion that the results of the study are valid only if LIDOCAINE is being used for intra-articular anesthesia of the distal interphalangeal joint. Most of the literature cited and used to contrast the results of the current study have used a long lasting local anesthetic solution (mepivacaine) which could have been a different effect under the described study set up/design.

---

## Round 0.2 · Minor Revisions

Thank you very much for revising your article in response to the reviewers' feedback. The two reviewers who reviewed your original submission have now reviewed this resubmission and both note that you have addressed their concerns. However, reviewer 2 asks again that you provide some images to illustrate your methodology and I agree that it would be very helpful if you could provide them. I do not anticipate that I will need to send your article back out review after this find round of edits.

·

Basic reporting

The authors have addressed all prior reviewer comments appropriately.

Experimental design

No comment

Validity of the findings

No comment

Additional comments

Thank you for making the suggested edits and clarifications.

Reviewer 2 ·

Basic reporting

No comment.

Experimental design

No comment.

Validity of the findings

No comment.

Additional comments

Dear authors, I am satisfied with the changed made to the revised manuscript, however as I suggested before including images or illustrations of the hoof defects created for pressure algometry would be useful to the final manuscript. I consider this point important to illustrate the manuscript and provide clarity.

---

## Round 0.3 · accepted · Accept

Thank you for the inclusion of the photographs showing your interventions; I agree with the reviewers that this is a helpful addition. It greatly adds to the clarity of your article. It is my pleasure to accept your article.